# Clinical Outcomes of SARS-CoV-2 Breakthrough Infections in Liver Transplant Recipients during the Omicron Wave

**DOI:** 10.3390/v15020297

**Published:** 2023-01-20

**Authors:** Anna Herting, Jacqueline Jahnke-Triankowski, Aenne Harberts, Golda M. Schaub, Marc Lütgehetmann, Darius F. Ruether, Lutz Fischer, Marylyn M. Addo, Ansgar W. Lohse, Julian Schulze zur Wiesch, Martina Sterneck

**Affiliations:** 1I. Department of Internal Medicine, University Medical Center Hamburg-Eppendorf, 20246 Hamburg, Germany; 2Department of Visceral Transplantation, University Medical Center Hamburg-Eppendorf, 20246 Hamburg, Germany; 3University Transplant Center, University Medical Center Hamburg-Eppendorf, 20246 Hamburg, Germany; 4German Center for Infection Research (DZIF), Partner Site Hamburg-Lübeck-Borstel-Riems, 38124 Braunschweig, Germany; 5Institute of Medical Microbiology, Virology and Hygiene, University Medical Center Hamburg-Eppendorf, 20246 Hamburg, Germany; 6Bernhard-Nocht-Institute for Tropical Medicine, Department for Clinical Immunology of Infectious Diseases, 20359 Hamburg, Germany; 7University Medical Center Hamburg-Eppendorf, Institute for Infection Research and Vaccine Development (IIRVD), 20246 Hamburg, Germany

**Keywords:** liver transplant recipients, SARS-CoV-2, COVID-19, VOC Omicron, post-acute COVID-19 syndrome

## Abstract

At the start of the pandemic, liver transplant recipients (LTR) were at high risk of developing severe COVID-19. Here, the outcomes of breakthrough infections in fully vaccinated LTR (*n* = 98) during the Omicron wave were assessed. In most patients, a mild disease course was observed, but 11 LTR (11.2%) required hospitalization for COVID-19-related complications. All patients survived. The LTR requiring hospitalization were older (67 years vs. 54 years; *p* < 0.001), had a higher Charlson comorbidity index (9 vs. 5; *p* < 0.001), and a lower anti-S RBD titer (Roche Elecsys) prior to infection (508.3 AU/mL vs. 2044 AU/mL; *p* = 0.03). Long-lasting symptoms for ≥4 weeks were reported by 37.5% of LTR (30/80). Risk factors in LTR included female sex (*p* = 0.01; Odds Ratio (OR) = 4.92 (95% confidence interval (CI) (1.5–16.5)) and dyspnea (*p* = 0.009; OR = 7.2 (95% CI (1.6–31.6)) during infection. Post-infection high anti-S RBD antibody levels were observed in LTR, and healthy controls (HC), while the cellular immune response, assessed by interferon-gamma release assay (EUROIMMUN), was significantly lower in LTR compared with HC (*p* < 0.001). In summary, in fully vaccinated LTR, SARS-CoV-2 breakthrough infections during the Omicron wave led to mild disease courses in the majority of patients and further boosted the humoral and cellular hybrid anti-SARS-CoV-2-directed immune response. While all patients survived, older and multimorbid LTR with low baseline antibody titers after vaccination still had a substantial risk for a disease course requiring hospitalization due to COVID-19-related complications.

## 1. Introduction

In the pre-vaccination era, COVID-19 mortality rates of up to 30% were reported in solid organ transplant recipients (SOT) [1]. Among liver transplant recipients (LTR), several studies showed increased morbidity and mortality compared to the general population [2]. In the pre-Omicron era, it has been discussed that not only the immunosuppression but the different patient comorbidities in this patient population were factors that determined worse outcomes [3,4,5,6]. Vaccinations against SARS-CoV-2 are estimated to have prevented 14.4 million deaths worldwide between December 2020 and December 2021 [7]. However, compared with the general population, LTR show a somewhat weaker humoral and cellular immune response after basic immunization as well as after booster vaccinations resulting in a lower seroconversion rate, lower median antibody titers, and a lower T cell response against spike proteins [8,9,10,11,12]. Therefore, despite high vaccination rates, and the high motivation of LTR to receive booster vaccinations, protection against severe COVID-19 courses might be less robust compared with the general population due to a weaker vaccination response [11]. Clinical features associated with poor vaccination responses in SOT include higher age, advanced renal insufficiency, diabetes, higher BMI, and arterial hypertension [8,9,11,13,14]. Additionally, immunosuppressive treatment regimens that include mycophenolate mofetil at higher dosages have been found to be risk factors for induction of a weaker vaccination response to COVID-19 vaccines in LTR [15,16,17].

Large cohort studies in the general population showed that the risk of severe outcomes, i.e., hospitalization or death, were lower for persons infected with the Omicron compared with patients infected with the Delta variant [18], even after adjusting for vaccination status and comorbidities, suggesting a reduced intrinsic severity of Omicron infections [19]. Since then, multiple Omicron subvariants have emerged, with BA.1, BA.2, and BA.4/5 accounting for almost all cases in Germany in the first half year of 2022. While a high rate of breakthrough infections has already been observed with the BA.1 subvariant [20], BA.4/5 variants seem to be even more transmissible due to greater immune evasion [21]. Whether the various subvariants lead to different clinical courses is not well known yet, and contradicting results were reported by different research groups [22,23,24]. There are only limited data available on the severity of the clinical course of COVID-19 [25], frequency of the development of long-lasting symptoms, and immune response in LTR with breakthrough infections during the Omicron wave.

Therefore, this single-center study from Northern Germany aimed to investigate the clinical course of a SARS-CoV-2 breakthrough infection and the consecutive immune response during the Omicron wave in fully vaccinated LTR.

## 2. Materials and Methods

At the Liver Transplant Center of the University Hamburg-Eppendorf, 675 adult LTR have been regularly followed up during the COVID-19 pandemic. In this study, LTR with a SARS-CoV-2 infection during the Omicron wave during the study period from 20 December 2021 to 31 July 2022 were included.

The study period and prevalence of the Omicron variants were based on local genome sequencing data by the Hamburg Surveillance project of the Leibniz Institute for Virology [26] and the Robert Koch Institute [27]. Since the beginning of 2022, the Omicron variants (in particular BA1; BA2; BA4/5) were the strains present in Hamburg/Germany with a prevalence of ≥99%.

In the participating LTR, SARS-CoV-2 infection was confirmed via a positive SARS-CoV-2 polymerase chain reaction (PCR) result or via an antigen test plus a positive nucleocapsid antibody test after the infection or via an antigen test plus typical symptoms plus a known source of infection. Participants were recruited during routine visits at the transplant center where SARS-CoV-2 vaccination and infection status was assessed regularly. Exclusion criteria were age < 18 years, lack of previous SARS-CoV-2 basic vaccination, pregnancy at the time of infection, and missing patient consent. Furthermore, 19 unselected healthy controls (HC) with a history of confirmed SARS-CoV-2 infection were recruited from hospital staff. Follow-up investigations and collection of blood samples were carried out during routine appointments 3 months (±8 weeks) post-infection. In addition, serum samples taken between the last vaccination and before the SARS-CoV-2 infection, that were stored at −20 °C, were tested. This study was approved by the local ethics committee of Hamburg, Germany (Reg. number PV7103 and PV7298), and all participants signed a written consent form.

Patient characteristics and clinical data were collected from the electronic medical records. Furthermore, all patients were asked to fill out a questionnaire to receive detailed information on the course and symptoms of their SARS-CoV-2 infection. Symptomatic participants were asked to rate their subjective COVID-19 disease severity on a scale from 1 (very mild) to 10 (very severe) in the questionnaire.

SARS-CoV-2 infected participants who were hospitalized were divided into two groups: (i) patients requiring hospitalization for COVID-19-related complications and (ii) patients with a SARS-CoV-2 infection or mild COVID-19 during hospitalization for non-COVID-19-related medical indications, clinical surveillance of COVID-19, or assessment in the emergency department. Furthermore, in patients requiring hospitalization for COVID-19-related respiratory distress, the National Institutes of Health (NIH) definition was used to categorize the disease severity in moderate (SpO_2_ ≥ 94%), severe (SpO_2_ < 94%), and critical (acute respiratory distress syndrome, septic or cardiac shock, exacerbation of underlying comorbidities) illness [28].

Based on the National Institute for Health and Care Excellence guidelines [29], ongoing symptomatic COVID-19 (osCOV-19) was defined as the persistence of acute COVID-19 symptoms for 4–12 weeks and post-COVID-19 syndrome (postCOV-19) as persisting symptoms >12 weeks after the onset of the infection.

### 2.1. Assessment of the Humoral and Cellular Spike-Specific Immune Response

Antibody levels were determined by the Roche Elecsys SARS-CoV-2 S assay in arbitrary units (AU) per ml as described previously [9] with a linear range from 0.4 AU/mL to 25,000 AU/mL. Samples that exceeded 25,000 AU/mL were manually diluted at 1:30 and retested. A negative test result was defined as <0.8 AU/mL, a low positive response between 0.8 AU/mL and 10^3^ AU/mL, and a positive response >10^3^ AU/mL. SARS-CoV-2 Nucleocapsid antibodies were assessed by the Elecsys anti-NC-SARS-CoV-2 Ig assay (Roche, Mannheim Germany; cutoff ≥ 1 COI/mL).

Cellular immune response was determined by a commercial spike-specific Interferon-Gamma-Release-Assay (IGRA, EUROIMMUN, Lübeck, Germany), as previously described [9]. The IGRA was interpreted according to the manufacturer’s instructions, with Interferon-gamma (IFN-γ) levels of <100 mlU/mL being negative, 100–200 mlU/mL being low positive, and >200 mlU/mL being high-positive [30].

### 2.2. Statistical Analysis

Descriptive statistics including median and interquartile range (IQR) for continuous variables or the number of patients and percentages for categorical variables were used to present epidemiological data. Differences between groups were assessed by either Pearson’s chi-squared test, Fisher’s exact test, Mann–Whitney U Test, or Wilcoxon-signed-rank test according to scales of measurement, sample size, and research question. A binomial logistic regression analysis was performed, including parameters that were significantly more common in our patients with long-lasting symptoms after COVID-19 and clinically relevant variables based on literature and background knowledge, to identify risk factors for developing osCOV-19 or postCOV-19 In order to avoid multicollinearity, we calculated the correlation coefficient r and applied a threshold of r < 0.85 for variables to be included in the final multivariate model with a prioritized selection of parameters that showed statistical significance [31]. A *p*-value < 0.05 was considered statistically significant. SPSS statistics (IBM Corp, Armonk, NY, USA) and Prism GraphPad Version 8.0.1 for Windows (Graph-Pad Software, San Diego, CA, USA) were applied for statistical analysis and to create graphs and figures.

## 3. Results

### 3.1. Study Cohort and Patient Characteristics

Altogether, in 98 of the 675 (14.5%) LTR cared for at the Liver Transplant Center, a laboratory-confirmed SARS-CoV-2 infection was diagnosed during the study period, based on either a positive PCR (*n* = 82) or a positive antigen test plus the development of a positive nucleocapsid antibody test after the infection (*n* = 11) or a positive antigen test plus typical symptoms plus a known source of infection (*n* = 4).

The characteristics of the 98 LTR included in the study are shown in Table 1. The majority of LTR had received three (60.2%, *n* = 59) or four (25.6%, *n* = 25) mRNA and/or vector-based vaccine doses prior to the infection, with a median time interval between infection and last vaccination of 130 days (IQR 88.8–183.3; *n* = 86). Vaccinations were offered according to the German guidelines and the majority of participants had received mRNA-based vaccines [9,11]. One LTR reported a previous SARS-CoV-2 infection with the Delta variant before the study period. Infected LTR had a median age of 56 years (IQR 42–65) and comorbidities, such as diabetes mellitus (DM) (*n* = 32; 32.7%), arterial hypertension (aHT) (*n* = 52; 53,1%), and renal insufficiency with an estimated glomerular filtration rate (eGFR) of <30 mL/min (*n* = 18 (18.4%), were common among the study group (Table 1). Median time since transplantation was seven years (IQR 3–13.3). Of the 68 patients who received a liver transplant at our institution in 2021 and 2022, sixteen fully vaccinated LTR (23.5%) developed a breakthrough infection in the first year after transplantation.

### 3.2. The Clinical Course of Omicron Breakthrough Infections in Fully Vaccinated LTR 

Altogether, 21 of 98 LTR were hospitalized during their SARS-CoV-2 infection, but only in 11 LTR hospitalization was indicated due to COVID-19-related complications (Appendix A), while the remaining ten patients (Appendix A) were either hospitalized for (a) other medical reasons (*n* = 6), (b) as a precaution to monitor their clinical disease course (*n* = 1), or (c) for assessment and/or administration of COVID-specific, antiviral treatment in the emergency room (*n* = 3). Overall, clinical courses not requiring hospitalization for severe COVID-19 were seen in 88.8% of LTR (*n* = 87). No patient in this cohort died from COVID-19 within the study period.

Nineteen patients (19.4%) (*n* = 13 not requiring and *n* = 6 requiring hospitalization) received COVID-19-specific medication including antiviral (*n* = 9; 9.2%), antibody treatment (*n* = 5; 5.1%), or a combination of both (*n* = 5; 5.1%) according to the treating physician´s or patient´s choice. Two LTR (2%) with a mild COVID-19 course after the breakthrough infection had received tixagevimab/cilgavimab as a prophylactic passive immunization due to known non-response to repeated COVID-19 vaccines.

### 3.3. The Clinical Course of LTR Requiring Hospitalization for COVID-19 Complications

Data on the 11 LTR who were hospitalized for COVID-19-related complications is shown in Appendix A. These LTR presented with pneumonia and respiratory distress (*n* = 5), acute kidney injury (AKI) due to COVD-19 related diarrhea or infection-related exsiccosis (*n* = 4), or a combination of both (*n* = 2). In addition to the AKI, one LTR presented with somnolence, and one had syncope. The median hospitalization duration of patients requiring hospitalization due to COVID-19 was 11 days (IQR 7–23.8) but the length of the hospital stay considerably varied. Two fully vaccinated LTR with COVID-19 breakthrough infections required intensive care management (Appendix A). Only six LTR received antiviral COVID-19-specific treatment (3 LTR Remdesivir; 2 LTR Sotrovimab; 1 LTR Tixagevimab/Cilgavimab), the other five LTR were admitted too late in the disease course, i.e., >5 days post onset of infection to be antivirally treated. In nine of the eleven LTR requiring hospitalization, pulmonary infiltrates were diagnosed by computer tomography (CT) and/or chest X-ray (CXR), seven LTR required supplemental oxygen. One LTR required assisted- and another LTR required invasive ventilation. Eight LTR developed a secondary infection, i.e., bacterial pneumonia or sepsis requiring a prolonged hospital stay. In LTR hospitalized with pneumonia (*n* = 7), we observed one, three, and two moderate, severe, and critical disease courses, respectively (Appendix A). For the remaining two LTR the available data were insufficient for reliable categorization.

The clinical characteristics of the 11 LTR requiring hospitalization for COVID-19 are compared with all the other LTR in Table 2. Of note, LTR requiring hospitalization were older (median age 67 y vs. 54 y; *p* < 0.001) and had more comorbidities (median CCI 9 vs. 5; *p* < 0.001). Therefore, aHT, DM, age > 60 y, eGFR < 30 mL/min, and obesity (body mass index (BMI) > 30 kg/m²) were significantly more frequently present than in the remaining patients (*p* = 0.008; 0.008; <0.001; 0.027; 0.037, respectively). Furthermore, patients requiring hospitalization had more often received a fourth (54.5% vs. 21.8 %; *p* = 0.029) or fifth vaccination (9.1% vs. 1.1%; *p* = 0.213) compared to the remaining LTR. However, as given in paragraph 3.5 in detail, despite the higher number of vaccinations the anti-S RBD antibody levels were significantly lower in LTR requiring hospitalization compared to the others. Of note, there was no significant difference concerning the vaccination scheme between LTR-requiring and not-requiring hospitalization. Moreover, we did not observe any differences regarding the immunosuppressive treatment between the two groups (Table 2). 

### 3.4. Range and Duration of Self-Reported Specific COVID-19 Symptoms after breakthrough infection 

In a subset of patients (*n* = 80), data on symptoms and the duration of COVID-19 were available through a multi-item questionnaire (Appendix A).

In most LTR, the assumed source of infection remained unclear (Appendix A), but nosocomial infections were rare (3.8%).

The median number of days until the first negative test result was 11 days (IQR 7.3–20.8; *n* = 52), but there were eight LTR with a known prolonged disease course defined by positive test results for >30 days (IQR 42.8 days-75.6 days). Three participants with prolonged SARS-CoV-2 positivity required hospitalization due to COVID-19-related complications. Although not statistically significant, LTR with prolonged SARS-CoV-2 positivity were more often treated with Mycophenolate Mofetil (62.5% vs. 30%; *p* = 0.109) or a combination of mTORi and MMF/Prednisone (25% vs. 3.4%; *p* = 0.052). Furthermore, these patients were more likely to have received no booster vaccination (37.5% vs. 10 %, *p* = 0.056). A comparison of clinical characteristics, vaccination status, and immunosuppression between patients with and without prolonged SARS-CoV-2 positivity is available in the Appendix A.

Cough, rhinorrhea, and fatigue were among the most common symptoms of LTR (Appendix A). Only four LTR reported an asymptomatic SARS-CoV-2 infection.

A total of 30/80 LTR (37.5%) suffered either from osCOV-19 or postCOV-19 after the acute COVID-19 breakthrough infection. Of the 30 LTR, 14 self-reported post-COVID-19 (post-COV-19) with persisting symptoms of at least twelve weeks (Figure 1A). Common symptoms included fatigue, cough, dyspnea, and difficulty concentrating (Figure 1B).

A comparison of LTR with and without long-lasting symptoms is shown in Table 3. A binomial logistic regression analysis was performed to identify risk factors associated with the development of long-lasting symptoms including all three parameters which were statistically more frequent in patients with osCOV-19 or postCOV-19 (Table 3) and clinically relevant variables based on the literature and background knowledge [32,33,34,35]. The resulting multivariate model was statistically significant with x² = 21.49; *p* = 0.003. The goodness of fit was assessed by the Hosmer–Lemeshow-Test, showing a good model fit with x² = 4.22 and *p* > 0.5. The sensitivity of our model was 66.7% and the specificity was 81.3%, with an overall classification accuracy of 75.6%. 

We found two variables in LTR that were significantly predictive of a later osCOV-19 or postCOV-19 status. Females (*p* = 0.01; Odds Ratio (OR) = 4.92 (95% confidence interval (CI) (1.5–16.5)) and LTR suffering from dyspnea during acute COVID-19 (*p* = 0.009; OR = 7.2 (95% CI (1.6–31.6)) were more likely to develop long-lasting symptoms. We observed no significant differences between osCOV-19 or postCOV-19 status and age (*p* = 0.65), CCI (*p* = 0.79), LTR with ≥3 vaccinations (*p* = 0.09), ≥3 immunosuppressive medications (*p* = 0.085), or difficulty concentrating (*p* = 0.429). All model coefficients and *p*-values are available in Table 4.

### 3.5. The Hybrid Cellular and Humoral Immune Response after Omicron Variant Breakthrough Infection in LTR

Detailed analysis of the humoral response after the SARS-CoV-2 infection revealed high anti-SARS-CoV-2 receptor-binding domain (anti-S RBD) levels in LTR (*n* = 58), and HC (*n* = 18) with a median antibody titer of 21,931 AU/mL, and 23,651 AU/mL (median sample collection post infection 90.5 d (IQR 62 d–111 d), 117 d (IQR 73 d–135 d)), respectively. Samples from patients who received monoclonal spike-antibody treatment during or before infection were excluded from the subsequent serological analysis. While all HC had antibody titers over 10^3^ AU/mL, in seven LTR, titers remained under 10^3^ AU/mL (Figure 2). Of note, the median anti-S RBD titer did not differ between LTR receiving versus those not-receiving MMF (Appendix A) nor between LTR receiving mono versus triple immunosuppression (Appendix A).

In 54 LTR, anti-S RBD antibody data were available after their respective last vaccination and before the SARS-CoV-2 infection. Three of the 54 LTR were vaccine non-responders, defined as having no detectable humoral response. The median antibody titer prior to infection was significantly lower in LTR requiring hospitalization for COVID-19 than in those who did not (508.3 AU/mL vs. 2044 AU/mL; *p* = 0.03; median sample collection before infection (35 d vs. 41 d; *p* = 0.844 (Figure 3). 

In 29 LTR, the antibody titers before and after infection were analyzed. Here, the infection led to a seven-fold increase in antibody titers (increase from 2689 AU/mL to 18,871 AU/mL; *p* < 0.001) (Figure 2E).

The spike-specific T-cell response was assessed by an IGRA in 37 LTR, and 19 HC, and the median sample collection time after infection was 96 d (IQR 80.5–110.5) and 117 d (IQR 73–135), respectively (Figure 2). IFN-γ levels were significantly higher in HC compared to LTR (898.5 mlU/mL vs. 3245 mlU/mL; *p* < 0.001). While all HC showed a high positive cellular response (>200 mlU/mL), four LTR remained low positive (100–200 mIU/mL and four LTR showed a negative (<100 mUI/mL) cellular response. While all HC had antibody titers over 10^3^ AU/mL and high IFN-γ levels, 26.5% (*n* = 9) of LTR had either a lower humoral or cellular immune response (Figure 2F).

## 4. Discussion

As the main result of this current single-center real-world observational study, we find that the majority of LTR with breakthrough SARS-CoV-2 infections and an adequate vaccination status endured rather mild disease courses during the Omicron wave. This demonstrates the high real-world efficacy of the COVID-19 vaccination in this patient population. However, 11.2% of LTR with breakthrough infections still required hospitalization for COVID-19-associated complications despite fulfilling the recommended vaccination status. These results are in line with mortality (2–10%) and hospitalization rates (3–56%) that were found in SOT [25,36,37,38] and a small Spanish single-center study that included 30 LTR with a hospitalization and mortality rate of 3.3% [25]. 

Importantly, our study was able to identify the particular risk factors associated with COVID-19 disease courses requiring hospitalization. More than 90% of the LTR with a need for hospitalization were older than 60 years and had multiple comorbidities (CCI 9 vs. 5; *p* < 0.001). Furthermore, all assessed risk factors for severe COVID-19 including aHT (*p* = 0.008), DM (*p* = 0.008), age < 60 y (*p* < 0.001), eGFR < 30 mL/min (*p* = 0.027), and BMI > 30 kg/m² (*p* = 0.037) were significantly more common in LTR requiring hospitalization than in those who did not. On the other hand, there was no difference in immunosuppressive treatment including the use of mycophenolate mofetil (MMF) or time since transplantation (Table 2). 

Respiratory distress due to pneumonia, often accompanied by bacterial superinfection, was the most common reason for hospitalization in our cohort. Of LTR requiring hospitalization due to COVID-19-related complications, three (27.3%) patients had a severe and two (18.2%) a critical COVID-19 disease course with assisted or invasive ventilation, according to the NIH guidelines. However, unlike in the general population, we also observed a high rate of AKI due to COVID-19-related diarrhea or exsiccosis leading to hospitalization. In accordance, high rates of gastrointestinal symptoms were also previously reported in LTR [3,5]. Patients requiring hospitalization due to COVID-19-related AKI were not categorized according to the NIH guidelines since this definition rather focuses on respiratory complications.

In addition to the 11 LTR requiring hospitalization for COVID-19-associated complications, 10 LTR were hospitalized with an asymptomatic SARS-CoV-2 infection or mild COVID-19 for clinical surveillance or other medical reasons. Therefore, the mere hospitalization rate which is often cited as a measure of the severity of COVID-19 in SOT probably overestimates true disease severity.

Furthermore, we assume that more stringent outpatient management of the LTR with breakthrough infections might have prevented the development of AKI and bacterial superinfections leading to hospitalizations in several cases, as would have the early initiation of antiviral treatment. In several published studies, the use of antiviral and early antibody treatment which showed excellent tolerability significantly reduced mortality, and the need for supplemental oxygen in SOT [39,40,41,42,43,44]. Therefore, early administration of antiviral treatment should be considered in LTR with comorbidities and a higher age who are thus at an increased risk for a severe disease course. However, the benefit of treatment in reducing hospital admissions and mortality in this particular patient population needs further investigation.

The clinical value of determining anti-S RBD antibody titers in SOT as a predictor for the risk of having a breakthrough infection and the risk of a severe clinical course in the case of a COVID-19 breakthrough infection is still under discussion, in particular during the Omicron pandemic. It has been shown by several groups that the neutralizing capacity of the Omicron variant by the spike-specific antibodies induced by vaccination with the so far available wildtype vaccines is quite poor [45,46] and only higher titers might offer protection [47]. Here, we also observed Omicron infections in LTR with high antibody levels prior to the infection (Figure 3). On the other hand, LTR requiring hospitalization had significantly lower antibody titers before infection than the remaining LTR (Table 2 and Figure 3A, B: 508.3 AU/mL vs. 2044 AU/mL, *p* = 0.03), although patients requiring hospitalization were more likely to have received a fourth or fifth vaccine dose compared with non-hospitalized patients (4th dose: 21.8% vs. 54.5%; *p* = 0.029; 5th dose: 1.1% vs. 9.1%; *p* = 0.213). This reflects the weak immune response to vaccination in LTR requiring hospitalization. Hospitalized LTR with COVID-19 were older and had more comorbidities than the LTR not-requiring hospitalization (Table 2). Old age and the presence of comorbidities are negative predictive factors for both a poor vaccination response and a severe disease course [4,9,11,12]. Therefore, although at our institution such high-risk LTR were primarily selected for a second and third booster early in autumn and winter 2021/2022, a severe disease course was not prevented in all cases. However, no patient in this cohort died of COVID-19. Our data on Omicron breakthrough infections in LTR are in agreement with the data of recent investigations including a large Danish cohort study that revealed that the vaccination protects against severe disease courses and mortality in most SOT [48]. However, we also conclude that additional booster vaccinations with Omicron-adapted vaccines should be considered, especially in LTR with low anti-S RBD antibody levels. 

Post-infection, the LTR in our cohort developed a robust humoral immune response. Anti-S RBD levels (21,931 vs. 1196 AU/mL) and IFN-γ levels in the IGRA (898.5 vs. 78.2 mlU/mL) were even higher in convalescent LTR than in LTR with four vaccinations from our previous cohort [11]. This is in accordance with the good protection against reinfection that we saw until the end of the follow-up of our study. However, how long the protection against reinfection prevails and whether it includes currently emerging Omicron variants like BQ1.1 and XBB.1 or future new variants remains uncertain. Furthermore, the cellular immune response was significantly weaker in LTR compared with HC, which can be attributed to the primarily T-cell-directed immunosuppressive therapy. Therefore, protection against severe disease courses after a subsequent re-infection might not be as solid or might wane faster over time than in healthy individuals.

An increasing public health problem may be the high number of patients affected by long-lasting COVID-19 symptoms [33,49]. Prevalence estimates of long-lasting symptoms range between 7.5 and 41% in the general population and 37.6% among hospitalized adults [50]. To our knowledge, data on the persistence of long-lasting symptoms in LTR during the Omicron wave has not been published to date. In our study, 30 of 80 LTR (37.5%) who completed the questionnaire reported ongoing symptoms for at least four weeks after the onset of infection. However, the percentage of participants with osCOV-19 could be overestimated, as data were only available via the questionnaire, which may have been preferentially filled out by individuals affected by ongoing symptoms. In line with other studies investigating the risk factors for long-lasting symptoms in the general population, female sex as well as experiencing dyspnea during acute COVID-19 were significantly associated with developing long-lasting symptoms in affected LTR [32,51], while age, vaccination status, ≥3 immunosuppressants, and comorbidities were not. Due to the small study population and the non-response bias, the explanatory power of our statistical analysis is limited and should be interpreted with caution. However, more research is needed, because of the potentially debilitating effects of long-lasting COVID-19 symptoms on health status and quality of life in this special patient population.

This is the first study from Germany describing the clinical outcome of a SARS-CoV-2 infection during the Omicron wave in a large cohort of LTR. Our study is an epidemiological, observational trial, thereby having some limitations. Patients were not randomized due to ethical reasons and there was no nucleocapsid testing for all 675 patients who were assessed for eligibility at the beginning and end of our study period. All LTR followed in our clinic were advised to get in contact with us in case of a SARS-CoV-2 infection. However, it is theoretically conceivable, that some patients with a mild or asymptomatic disease course have not informed us of their infection, thereby this study might underestimate the number of mild infections. Furthermore, data on symptoms and duration of COVID-19 were assessed through a questionnaire, which was not returned by all study participants (participants who returned the questionnaire *n* = 80; 81.6%) and therefore may have led to a non-response bias. Moreover, we were not able to sequence for Omicron-specific mutations, since only a minority of the samples were tested at our institution. However, genome sequencing data on the spread of the different SARS-CoV-2 variants on a local level is available through the Hamburg Surveillance project by the Leibniz Institute for Virology [26]. Due to the lack of variant sequencing in our cohort, we were unable to analyze potential differences between clinical course or immune response in individuals infected with the different Omicron subvariants. Since not all patients tested themselves repeatedly after their SARS-CoV-2 infection and due to the small number of LTR with known prolonged SARS-CoV-2 positivity, the explanatory power of the statistical analysis is limited. Ultimately, due to the small number of hospitalized study participants, we were unable to perform a multivariate analysis for the LTR requiring hospitalization.

In summary, in fully vaccinated LTR, SARS-CoV-2 breakthrough infections during the Omicron wave led to mild disease courses in the majority of patients and further boosted the humoral and cellular hybrid anti-SARS-CoV-2-directed immune response. While all patients survived, older and multimorbid LTR with low baseline antibody titers after vaccination still had a substantial risk for a disease course requiring hospitalization due to COVID-19-related complications. Therefore, providers and patients should be educated about the increased risk for severe disease courses and further booster vaccinations with variant-adapted vaccines should be considered. LTR with multiple comorbidities and higher ages should be preferentially assessed for booster vaccinations and the early administration of antiviral treatment to prevent hospital admissions in the case of a breakthrough infection.

## Figures and Tables

**Figure 1 viruses-15-00297-f001:**
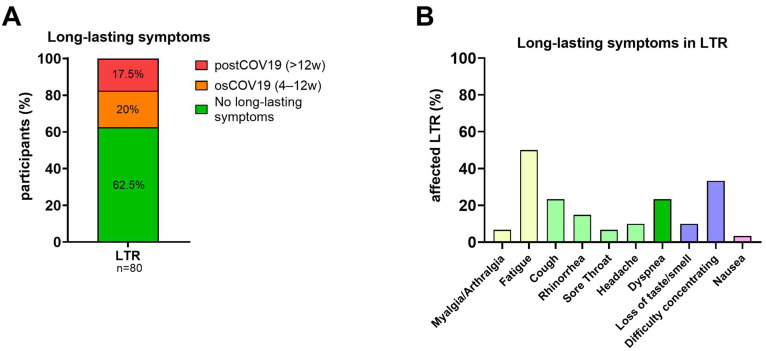
Symptoms and duration of SARS-CoV-2 breakthrough infections. (**A**): Percentage of LTR (*n* = 80) reporting long-lasting symptoms: subdivided into osCOV-19 (4–12 weeks) and postCOV-19 (>12 weeks). (**B**): Reported long-lasting symptoms in LTR (*n* = 30). Abbreviations: osCOV-19: ongoing symptomatic COVID-19; postCOV-19: post-COVID-19 syndrome; w: weeks.

**Figure 2 viruses-15-00297-f002:**
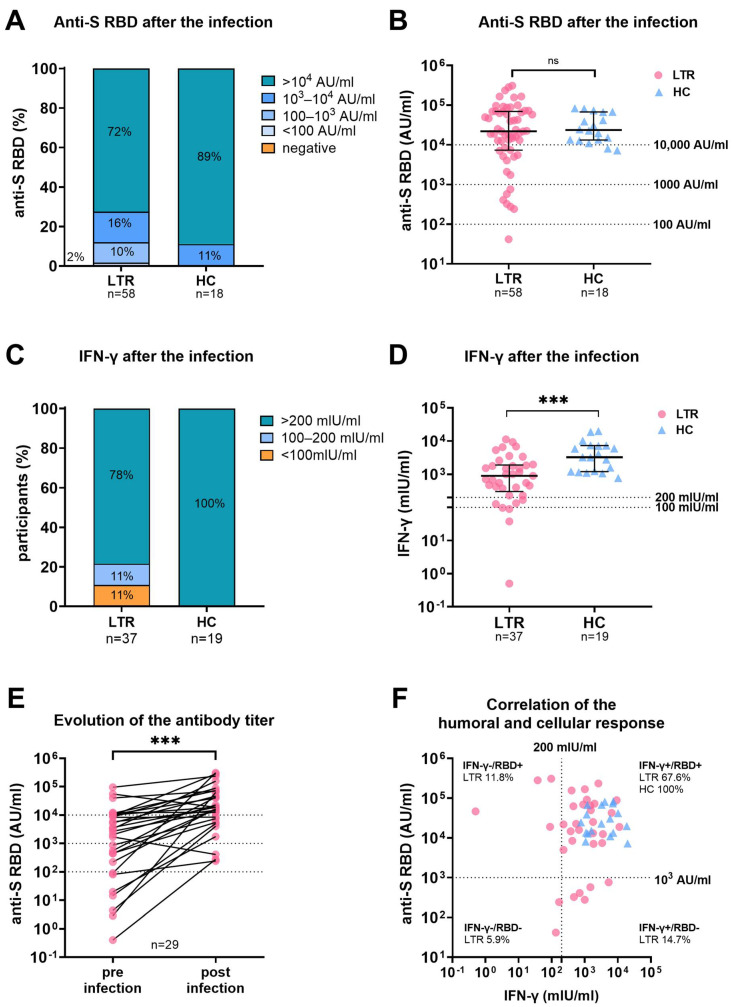
Humoral and cellular immune response in LTR and HC after SARS-CoV-2 infection. (**A**): Percentages of anti-S RBD titers in LTR, and HC between <0.4, <100, 100–10^3^, 10^3^–10^5^, and >10^5^ Arbitrary Units (AU)/mL. (**B**): Scattergram presenting individual anti-S RBD levels of LTR, and HC. (**C**): Percentages of LTR, and HC showing a negative (<100 mlU/mL), low positive (100–200 mlU/mL,) and high positive (>200 mlU/mL) Spike-specific T cell response as measured by IFN-γ release. (**D**): Scattergram with individual interferon-gamma (IFN-γ) levels afterSARS-CoV-2 breakthrough infection. (**E**): Evolution of anti-S RBD levels in LTR for whom samples were available before and after SARS-CoV-2 infection. (**F**): Correlation of humoral and cellular immune response in LTR (*n* = 37) and HC (*n* = 18). Dotted lines indicate cut-off values for IFN-γ (>200 mlU/mL) and anti-S-RBD (>10^3^ AU/mL) levels. Statistical Analysis was performed by Mann–Whitney Test (B, D) or Wilcoxon-signed-rank-test (E). Dotted lines indicate previously defined cut-off values. Dots and triangles were used to represent LTR, and HC, respectively. Ns and *** represent *p*-value > 0.05 and *p*-value < 0.0005, respectively.

**Figure 3 viruses-15-00297-f003:**
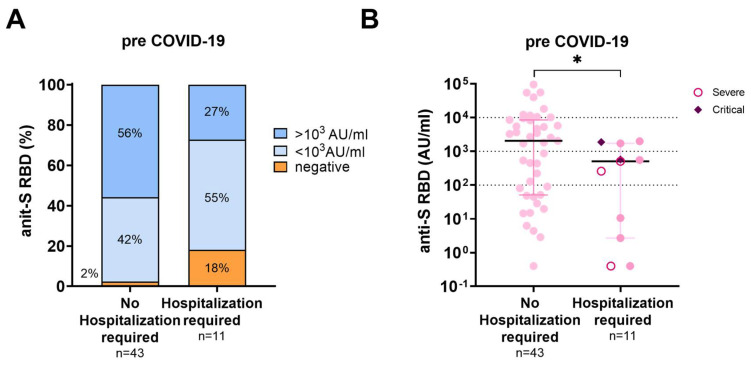
Humoral immune response prior to SARS-CoV-2 infection according to disease severity. (**A**): Percentages of anti-S RBD titers pre-COVID-19 in LTR with COVID-19 breakthrough infections requiring hospitalization and those not-requiring hospitalization. (**B**): Scattergram of individual anti-S RBD titers in LTR with COVID-19 breakthrough infections requiring hospitalization and those not requiring hospitalization. Patients with severe disease (*n* = 3) according to the NIH guidelines are presented by an open circle and patients with critical disease (*n* = 2) by a diamond. Statistical Analysis was performed by Mann–Whitney Test, * represents a *p*-value below <0.05.

**Table 1 viruses-15-00297-t001:** Baseline Characteristics.

Characteristics ^1^	LTR *n* = 98 *n* (%)/Median (IQR)	HC *n* = 19 *n* (%)/Median (IQR)
Age at the time of infection (years)	56 (42–65)	32 (25–47)
Females	46 (46.9)	11 (57.9)
BMI (kg/m^2^)	24 (21.5–26.9)	-
Time since transplantation (years)	7 (3–13.3)	-
**Etiology of liver disease**		
Alcoholic liver disease	14 (14.3)	-
Autoimmune	25 (25.5)	-
Viral	15 (15.3)	-
Hepatocellular carcinoma	8 (8.2)	-
Other	36 (36.7)	-
**Risk factors**		
Diabetes	32 (32.7)	0
Arterial Hypertension	52 (53.1)	2 (10.5)
Age > 60 years	37 (37.8)	4 (21.1)
eGFR < 30 mL/min	18 (18.4)	-
BMI > 30 kg/m^2^	13 (13.3)	1 (5.3)
2 Risk factors	46 (46.9)	-
≥3 Risk factors	21 (21.4)	-
Charlson comorbidity index	5 (4–8)	-
**Vaccination status**		
Second dose	12 (12.2)	0
Third dose	59 (60.2)	15 (78.9)
Fourth dose	25 (25.6)	4 (21.1)
Fifth dose	2 (2)	0
Time between last vaccine and infection (days)	130 (88.8–183.3) (*n* = 86)	139 (96.5–179) (*n* = 17)
**Immunosuppression**		
Tacrolimus	17 (17.3)	-
Cyclosporine	3 (3.1)	-
mTORi	2 (2)	-
CNI + MMF	24 (24.5)	-
CNI + AZA	3 (3.1)	-
CNI + mTORi	12 (12.3)	-
CNI + prednisone	16 (16.3)	-
mTORi + MMF	1 (1)	-
mTORi + AZA	0	-
mTORi + prednisone	4 (4.1)	-
≥3 Immunosuppressants	16 (16.3)	-
**Laboratory values**		
Leucocytes (Mrd/L)	5.6 (4.1–7.3) (*n* = 84)	-
Lymphocytes (Mrd/L)	1.3 (0.8–2) (*n* = 77)	-
eGFR (ml/min)	62 (33.8–89.8)	-
**Management of SARS-CoV-2 infection**		
Outpatient	77 (78.6)	19 (100)
Non-COVID related hospitalization	10 (10.2)	0
COVID-related hospitalization	11 (11.2)	0

^1^ Baseline Characteristics of all study participants. Frequencies and percentages (*n* = (%)) are given for nominal and ordinal variables. If no “*n*” value is given, data were available for all study participants. For numerical variables median and interquartile range (median (IQR)) were calculated. Abbreviations: LTR: liver transplant recipients; HC: healthy controls; IQR: Interquartile range; BMI: Body Mass Index; eGFR: estimated glomerular filtration rate; mTORi: mTOR inhibitor; CNI: Calcineurin inhibitors; AZA: Azathioprine.

**Table 2 viruses-15-00297-t002:** Vaccinated liver transplant recipients with COVID-19 breakthrough infections requiring hospitalization.

Characteristics ^1^	LTR Not Requiring Hospitalization *n* = 87 *n* (%)/Median (IQR)	LTR Requiring Hospitalization *n* = 11 *n* (%)/Median (IQR)	*p*-Value
Age at the time of infection (years)	54 (40–63)	67 (65–71)	<0.001
Female	41 (47.1)	5 (45.5)	0.818
BMI (kg/m²)	23.7 (21.2–26.6)	26.5 (24.9–31.1)	0.004
Time since transplantation (years)	7 (3–13)	11 (1–14)	0.831
**Risk factors**			
Diabetes	24 (27.6)	8 (72.7)	0.008
Arterial hypertension	42 (48.3)	10 (90.9)	0.008
Age > 60 years	27 (31)	10 (90.9)	<0.001
eGFR < 30 mL/min	13 (14.9)	5 (45.5)	0.027
BMI > 30 kg/m²	9 (10.3)	4 (36.4)	0.037
2 Risk factors	36 (41.4)	10 (90.9)	0.002
≥3 Risk factors	11 (12.6)	10 (90.9)	<0.001
Charlson comorbidity index	5 (4–7)	9 (7–14)	<0.001
**Vaccination status**			
Second dose	11 (12.6)	1 (9.1)	1.0
Third dose	56 (64.4)	3 (27.3)	0.024
Fourth dose	19 (21.8)	6 (54.5)	0.029
Fifth dose	1 (1.1)	1 (9.1)	0.213
Homologous mRNA-based vaccination	72 (85.7) (*n* = 84)	9 (90) (*n* = 10)	1.0
Heterologous mRNA/vector-based vaccination	12 (14.3) (*n* = 84)	1 (10) (*n* = 10)	1.0
**Anti-S RBD antibody titer pre-infection**			
Median anti-S RBD (AU/mL)	2044 (*n* = 43)	508.3 (*n* = 11)	0.03
Anti-S RBD < 10^3^ AU/mL	19 (44.2) (*n* = 43)	8 (72.7) (*n* = 11)	0.175
**Immunosuppression**			
Monotherapy	20 (23)	2 (18.2)	1.0
CNI + MMF /mTORi/Prednisone/AZA	49 (56.3)	6 (54.5)	0.911
mTORi + MMF/Prednisone/AZA	3 (3.4)	2 (18.2)	0.095
Additional MMF medication	28 (32.2)	4 (36.4)	0.746
≥3 Immunosuppressants	15 (17.2)	1 (9.1)	0.686
**COVID-19 therapy**			
Antiviral	6 (6.9)	3 (27.3)	-
Antibody	2 (2.3)	3 (27.3)	-
Combination of both	5 (5.7)	0	-
Dexamethasone	-	4 (36.4)	-
Reduction of Immunosuppressants	-	5 (45.5)	-
**Laboratory values**			
Leucocytes (Mrd/L)	5.8 (4.1–7.3) (*n* = 73)	4.6 (3.5–6.9) (*n* = 11)	0.195
Lymphocytes (Mrd/L)	1.5 (0.8–2) (*n* = 66)	0.9 (0.8–1.4) (*n* = 11)	0.238
eGFR (ml/min)	65 (36–93)	32 (18–50)	0.015

^1^ Comparison of LTR requiring hospitalization due to COVID-19-related complications with those who did not. Frequencies and percentages (*n* = (%)) are given for nominal and ordinal variables. For numerical variables median and interquartile range (median (IQR)) were calculated. If no “*n*” value is shown, data were available for all study participants. Statistical analysis was performed with Pearson’s chi-squared Test, Fisher’s exact test, or Mann–Whitney U test. Abbreviations: anti-S RBD: anti-SARS-CoV-2 receptor-binding domain.

**Table 3 viruses-15-00297-t003:** Liver transplant recipients with long-lasting symptoms.

Characteristics ^1^	LTR with Long-Lasting Symptoms *n* = 30 *n* (%)/Median (IQR)	LTR without Long-Lasting Symptoms *n* = 50 *n* (%)/Median (IQR)	*p*-Value
Age at the time of infection (years)	57 (39.5–65.5)	54.5 (42.8–65.3)	0.876
Female	17 (56.7)	17 (34)	0.047
BMI (kg/m²)	24.1 (21.4–28.1)	23.8 (21.1–26.6)	0.522
Time since transplantation (years)	5.5 (2.8–11)	7 (3–14)	0.551
**Risk factors**			
Diabetes	10 (33.3)	15 (30)	0.755
Arterial hypertension	14 (46.7)	27 (54)	0.525
Age > 60 years	13 (43.3)	16 (32)	0.307
eGFR < 30 mL/min	4 (13.3)	10 (20)	0.447
BMI > 30 kg/m^2^	5 (16.7)	4 (8)	0.284
2 Risk factors	14 (46.7)	23 (46)	0.954
≥3 Risk factors	5 (16.7)	9 (18)	0.879
Charlson comorbidity index	5 (4–7.3)	5 (4–8)	0.946
**Vaccination status**			
Second dose	2 (16.7)	7 (14)	0.471
Third dose	18 (60)	34 (68)	0.468
Fourth dose	9 (30)	8 (16)	0.138
Fifth dose	1 (3.3)	1 (2)	1.0
Immunosuppression			
Monotherapy	8 (26.7)	8 (16)	0.248
CNI + MMF /mTORi/Prednisone/AZA	18 (60)	31 (62)	0.859
mTORi + MMF /Prednisone/AZA	1 (3.3)	2 (4)	1.0
≥3 Immunosuppressants	3 (10)	9 (18)	0.52
Additional MMF medication	9 (30)	19 (38)	0.468
**Humoral immune response**			
Anti-S RBD prior to infection (AU/mL)	3666 (388.6–15,185) (*n* = 17)	1715 (11.6–5441) (*n* = 24)	0.058
Anti-S RBD post-infection (AU/mL)	19,868 (13,122–72,390) (*n* = 19)	22,016 (6110–63,960) (*n* = 33)	0.665
**Symptoms of acute COVID-19**			
Fever	14 (53.3)	16 (33.3)	0.239
Myalgia/arthralgia	13 (43.3)	15 (31.3)	0.279
Fatigue	22 (73.3)	28 (58.3)	0.179
Cough	17 (56.7)	26 (54.2)	0.829
Rhinorrhea	18 (60)	26 (54.2)	0.613
Sore throat	20 (66.7)	22 (45.8)	0.073
Headache	15 (50)	17 (37)	0.26
Nausea	4 (13.3)	3 (6.3)	0.287
Diarrhea	5 (16.7)	7 (14.6)	1.0
Dyspnea	12 (40)	6 (12.5)	0.005
Loss of smell/taste	5 (16.7)	3 (6.4)	0.25
Difficulty concentrating	12 (40)	7 (14.6)	0.011
**COVID-19 disease course**			
SARS-CoV-2 positivity < 30 days	4 (13.3)	3 (6)	0.416
Hospitalization for COVID-19 required	4 (13.3)	3 (6)	0.416
**Laboratory values**			
Leucocytes (Mrd/L)	5.8 (3.9–7.3) (*n* = 26)	5.4 (3.9–7.2)(*n* = 41)	0.607
Lymphocytes (Mrd/L)	1.5 (0.9–1.8) (*n* = 26)	1.3 (0.7–2) (*n* = 34)	0.493
eGFR (ml/min)	62.5 (38.3–92.3)	60.5 (31.8–90)	0.702

^1^ Description and comparison of LTR reporting long-lasting symptoms for more than 4 weeks and those who did not report persisting symptoms. Frequencies and percentages (*n* = (%)) are given for nominal and ordinal variables. If no “*n*” value is given, data were available for all study participants. For numerical variables median and interquartile range (median (IQR)) were calculated. Statistical analysis was performed with Pearson’s chi-squared Test, Fisher’s Exact Test, or Mann–Whitney U Test.

**Table 4 viruses-15-00297-t004:** Risk factors for long-lasting COVID-19 in LTR.

	Regression Coefficient	*p*-Value	Odds Ratio	95% Confidence Interval
Female sex	1.593	0.01	4.917	1.469–16.461
Age (years)	0.011	0.651	1.011	0.963–1.062
Charlson comorbidity index	−0.036	0.799	0.965	0.733–1.270
≥3 immunosuppressants	−1.419	0.085	0.242	0.048–1.213
Difficulty concentrating	0.553	0.429	1.738	0.441–6.852
Dyspnea	1.977	0.009	7.224	1.649–31.644
≥3 vaccine doses	1.079	0.085	2.942	0.842–10.278

A multivariate binomial logistic regression analysis was performed including clinically and statistically relevant predictor variables.

## Data Availability

Individual participant data will not be shared.

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
