# Peer review of "Clinical Outcomes of SARS-CoV-2 Breakthrough Infections in Liver Transplant Recipients during the Omicron Wave"

_viruses, 2023, doi:10.3390/v15020297_

Round 1

Reviewer 1 Report

The introduction should stand by itself without needing the abstract to introduce into the context. The first sentence should mention the context of covid, although it becomes obvious later. 

This is a very interesting single liver transplant centre real life experience reflecting Covid severity in adult liver transplant recipients and waiting list patients during the Omicron variant phase of the SARS-CoV-2 pandemic and may lead to conclusions of vaccination policies for adult liver transplant recipients and chronic liver failure patients and vaccination efficiency for the future. These aspects were very unclear at the time and let to huge level of insecurity around vaccination advice and for this particular group of patients, which in our centre had led to multiple consecutive communication initiatives with the respective patient groups. 

Round 2

Reviewer 2 Report

The authors addressed all remarks with detailed description and provided further analysis. I have no more suggestions and appreciate the authors´ thorough revision.

One minor: Line 24 (Abstract): liver transplant recipient should be spelled out once before using "LTR"